# GAN-based medical image small region forgery detection via a two-stage cascade framework

**Jianyi Zhang** [1,2]☯*, **Xuanxi Huang**[1]☯, **Yaqi Liu**[1], **Yuyang Han**[1], **Zixiao Xiang**[1]

**1** Beijing Electronic Science and Technology Institute, Beijing, China, **2** University of Louisiana at Lafayette, Lafayette, Louisiana, United States of America

☯ These authors contributed equally to this work.
* zjy@besti.edu.cn

## Abstract

Using generative adversarial network (GAN) Goodfellow et al. (2014) for data enhancement of medical images is significantly helpful for many computer-aided diagnosis (CAD) tasks. A new GAN-based automated tampering attack, like CT-GAN Mirsky et al. (2019), has emerged. It can inject or remove lung cancer lesions to CT scans. Because the tampering region may even account for less than 1% of the original image, even state-of-the-art methods are challenging to detect the traces of such tampering. This paper proposes a two-stage cascade framework to detect GAN-based medical image small region forgery like CT-GAN. In the local detection stage, we train the detector network with small sub-images so that interference information in authentic regions will not affect the detector. We use depthwise separable convolution and residual networks to prevent the detector from over-fitting and enhance the ability to find forged regions through the attention mechanism. The detection results of all sub-images in the same image will be combined into a heatmap. In the global classification stage, using gray-level co-occurrence matrix (GLCM) can better extract features of the heatmap. Because the shape and size of the tampered region are uncertain, we use hyperplanes in an infinite-dimensional space for classification. Our method can classify whether a CT image has been tampered and locate the tampered position. Sufficient experiments show that our method can achieve excellent performance than the state-of-the-art detection methods.

## 1 Introduction

Due to the privacy of medical images, the lack of data has always been a significant problem for machine learning tasks related to medical images. One way to solve this problem is the generative adversarial network (GAN) [1], which can generate images that are highly similar to real images. GAN has been widely concerned in the medical image field. Several studies have used GAN to generate medical images for data enhancement and achieved gratifying performance. The image quality generated by GAN is enough to confuse radiologists. Therefore, once this technology is used for malicious attacks, it will lead to serious consequences.

**Data Availability Statement:** All the code and data can be found here: https://github.com/BESTICSP/CT-GAN-Detector.

**Funding:** This work is supported by the Fundamental Research Funds for the Central

Universities (328202204). There was no additional external funding received for this study. The funders had no role in study design, data collection and analysis, decision to publish, or preparation of the manuscript.

**Competing interests:** The authors have declared that no competing interests exist.

Using the deep convolution neural network can detect the GAN-generated image [2–4]. Moreover, the detection accuracy can be improved through feature engineering [5–10]. To the knowledge of this paper, there is no detection method for GAN forged medical images. Although there is no specific solution to detect the medical images generated by GAN, there are some domain generic methods. For example, Frank *et al.* used discrete cosine transform (DCT) to detect GAN generated images [9]. Marra *et al.*, through incremental learning, can detect new GAN-generated images with only a small number of samples [11]. Cozzolino *et al.* learn feature extraction through auto-encoders and generalize the model through a small number of samples [12].

CT-GAN [13], a GAN difficult to detect even with state-of-the-art methods, emerged. It can inject or remove large lung nodules from CT images. Examples of CT-GAN inject/remove tampering of lung nodules are shown in Fig 1. The number of large lung nodules is a significant marker of lung cancer. Therefore, CT-GAN can make doctors misjudge the patient's condition, seriously threatening the patient's life safety. In addition, this attack may also be used to defraud medical insurance and maliciously discredit competitors.

However, new attacks like CT-GAN challenge the current detection methods. This type of attack is a GAN-based automated 3D tampering attack. It generates only a minimal area, and the surrounding area is used as a constraint condition to train a conditional generative

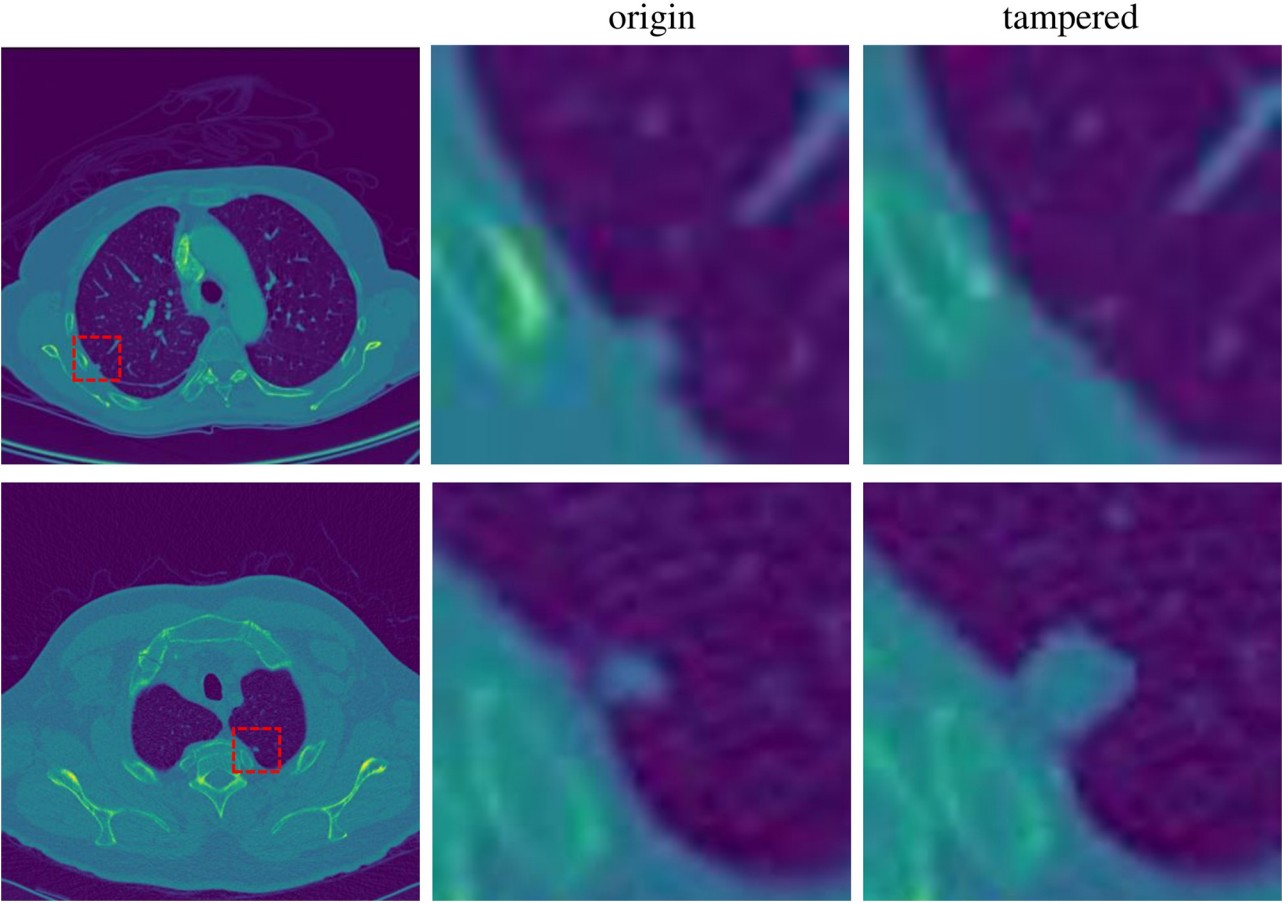

**Fig 1. CT-GAN tampered samples.** The first row shows the removal tampering of CT-GAN. A lung nodule is removed from the CT slice image by CT-GAN. The second row shows the injection tampering of CT-GAN. A small nodule was tampered with as a large nodule by CT-GAN.

adversarial network (CGAN). In that case, the generated image will be closer to the real image. We call the attack using CGAN to forge a very small region in an image as a GAN-based small region forgery attack. At present, no solution can effectively detect the GAN-based small region forgery attack in medical images. The characteristic of the attack is that the ratio of the region generated by GAN is very small. Although some methods, such as Rössler *et al.* [3], can detect partial generation, such as face manipulation. However, because medical images' style, content, and storage format are very different from the normal images and the tampered region is too small, even state-of-the-art detection methods can not effectively detect GAN-based small region forgery attacks in medical images. It is conceivable that our medical image security is facing a considerable threat.

In order to solve the above-mentioned problem, we propose a novel cascade framework based on a local detection network and a global classification method that can detect GAN-based small region forgery attacks in medical images. The first stage is local detection. We crop a small sub-image from the CT slice image to train the detector network. The sub-image size is small enough so that interference information in authentic regions will not affect the detector. Because the training data only has a single channel and the training data size is small, it is easy to over-fit. Therefore, we design a lightweight neural network with fewer parameters and use early stopping to prevent over-fitting. After training, the detector can detect the tampered region effectively. Then, we traverse the entire CT slice image by sub-image. The detection result of all sub-images will be combined and output as a heatmap. It can indicate which region may be tampered. The second stage is global classification. Since CT-GAN can adjust the size of the tampered region to a certain extent, we use gray-level co-occurrence matrix (GLCM) features to train principal component analysis (PCA), and classifier models for global classification. Compared with the method that uses the whole image as input, this method can locate the tampered coordinate and requires less training data but has faster training speed and higher accuracy.

The main contributions are as follows:

- We propose a novel cascade framework based on local detection and a global classification to detect and locate the tampering regions caused by GAN-based automated 3D medical imagery attacks, including injection and removal.

- For detection of GAN-based and small region forgery, we propose a local detection network with channel attention, spatial attention, depthwise separable convolution, and residual networks. It can better find information on small areas in the image and prevent over-fitting.

- For detection of the realistic alters that consider correlations between scans, we design a global classification method base on hyperplanes in an infinite-dimensional space with the gray-level co-occurrence matrix (GLCM) as input features. It can effectively cooperate with local detection to classify medical images.

- Experiments show that, for GAN-based small region forgery attacks in the medical image like CT-GAN, our method can achieve excellent performance.

For the reproducibility of the proposed method, we have published our source code online at https://github.com/BESTICSP/CT-GAN-Detector.

The rest of this paper is organized as follows. In Section 2, we discussed the background and related work of the detection of GAN generated images in recent years. Moreover, in Section 3, we explained our method in detail. Furthermore, in Section 4, we describe our experimental results. In Section 5, we discuss our method, and in Section 6 we draw our conclusions.

## 2 Background and related works

### 2.1 Medical image

The medical image uses some particular medium to interact with the human body to show the structure of the internal tissues or organs of the human. Digital imaging and communications in medicine (DICOM) is an international standard for medical images and their related information. It is widely used in various radiological diagnostic equipment (X-ray, CT, MR, ultrasound, etc.). All medical images of patients are stored in DICOM file format. The data used in this paper are mainly CT images with DICOM format. CT equipment scans slices one after another around a certain part of the patient's body. A complete CT scan may include about 300 slice images. The scanned image is multi-layered. A three-dimensional image can be formed by stacking layers of slice images on the z-axis.

The definition of medical images such as CT is positively correlated with radiation dose. In contrast, high-dose radiation may damage patient's health, so it is difficult to improve the definition of medical images. Besides, medical images have only one channel. So GAN can easily fit the distribution of medical images than normal three-channel color images.

### 2.2 Generative adversarial network

Since GAN was proposed by Goodfellow *et al.*, it has been one of the hot spots in the computer vision (CV) field. The GAN model is different from the traditional neural network structure. GAN includes a generative model $G$ and a discriminative model $D$. $G$ generates a new sample from random noise, and $D$ distinguishes whether the input sample is a real sample. The task of $G$ is to generate images that $D$ cannot distinguish. At the same time, the task of $D$ is to distinguish between the images generated by $G$ and the real images. The two networks compete against each other during training through this min-max game. In this way, $G$ can learn the data distribution of the real sample. Up to now, GAN has derived a large number of variants, such as WGAN [14], PGGAN [15], StyleGAN [16] and so on. These variants are widely used in various CV tasks.

### 2.3 Application of GAN in medical image

Medical images are different from normal images and have robust privacy. Even though there are many public data sets such as LIDC-IDRI, DDSM MIAS, OASIS, etc., the medical data sets are still insufficient. Because GAN can effectively alleviate the lack of training data, there are also a large number of researches of medical imaging using GAN. In recent years, the more frequently used GAN variants in medical imaging are pix2pix [17] and CycleGAN [18]. GAN is widely used in image synthesis [19–24], noise reduction [25–28], cross-modality [29–34], image enhancement [35], image super-resolution [36, 37], image segmentation [19], and many other aspects, providing significant help for the computer-aided diagnosis (CAD).

### 2.4 Detect the GAN-generated image

Because of the high performance of GAN, it has gradually become a trend to use deep learning to distinguish whether an image is generated by GAN. Due to the excellent performance of convolutional neural networks (CNN) in CV tasks, CNNs, such as ResNet [38], XceptionNet [39], and EfficientNet [40], are widely used in various CV fields, including digital image forensics [2, 4, 6]. Besides, Andreas *et al.* [3] prove the superior performance of XceptionNet in image source detection.

Using some features can make the network perform better. A way to distinguish whether an image is generated by GAN is to use GAN fingerprint. [7]. GAN will leave special

fingerprints in the generated image due to its structure. Through deep learning, learn those fingerprints as a feature. Then it can be used to distinguish the source of the image. Some people use the shortcomings of GAN to find some special features to better distinguish whether a image is generated by GAN. For example, McCloskey and Albright find that the saturated or underexposed pixels of image will be suppressed by the normalization operation of the GAN generator [5], and use this feature to distinguish the real camera images and GAN images. Because the statistical characteristics of GAN images are different from real images, some people use three co-occurrence matrices on RGB channels as features to distinguish the source of the image [8]. Zhang *et al.* suppress the image content information by converting the image to the YCrCb color space and then use the Scharr operator and the gray-level co-occurrence matrix(GLCM) to obtain edge features, allowing them to simultaneously detect GAN images and copy-move images [6]. In addition, someone distinguishes the source of the image from the defects of up-sampling operations in GAN. Frank *et al.* found that the up-sampling in GAN will cause grid-like artifacts in the generated images after DCT operation [9], which can be used to distinguish the source of the image. Durall *et al.* found that the images generated by GAN cannot reproduce the actual spectral distribution [10], which is also due to the upsampling operation. Therefore, after using azimuthal integration to extract the spectral features, using SVM or K-Means can distinguish the source of the image without the need to train a deep CNN.

## 2.5 Challenge

The CT-GAN paper also proposed some detection methods that may be useful. Unfortunately, these methods are not suitable for GAN-based small region forgery attack in medical image like CT-GAN. The reasons are as follows.

On the one hand, there is a huge difference between medical images and normal images. Medical images show the structure and density of human internal tissues or organs, so they have unique content and style. Medical images, such as CT, MR, X-ray, etc., are all taken with special equipment different from general photographing equipment and are saved according to the DICOM standard. Medical images are all single-channel in terms of the image data format, and the pixel values range of the medical image is about 4096. Compared with the normal gray-scale image ranging from 0 to 255, the range of pixel values of medical images is 16 times larger. Therefore, the pre-training model of normal images only has little effect on medical images. In addition, methods that need to extract features from three channels of an image, such as [6] that needs to compare three features extract from different channels, are not appropriate. The co-occurrence matrix is one of the most effective features to distinguish whether an image is generated by GAN. However, due to the expansion of the pixel value range, the cost of calculating the co-occurrence matrix will increase to unacceptable. So the method using the co-occurrence matrix [8] cannot work too. We did a detailed analysis on the countermeasures that are listed in [41]. Unfortunately, all of them failed to detect or prevent the nodule forgery for chest CT.

On the other hand, GAN-based small region forgery attacks are more difficult to detect. Take CT-GAN as an example. CT-GAN is a 3D CGAN. It designs a 3D network that references the pix2pix structure. The generator of CT-GAN is a 3D UNet [42] structure. It cuts out a small cuboid from a series of CT slices of the patient, then scales it into a small cube of $32^3$ pixels and masks the $16^3$ pixels in the center of the cube to zero. This cube with a masked center is input into the generator as a condition. CT-GAN trained two models. Those models can generate large or small nodules in the cube's center. It is worth mentioning that the size of a CT image is ($512 \times 512$) pixels, in which the region modified by CT-GAN is less than

(32 × 32). That means the minimum number of pixels that have been tampered with is only 1/1024 of the total. Fig 2 shows a CT image injected into a lung nodule.

Hence, as we can see from Figs 1 and 2, Each tampering operation by CT-GAN will modify 1/1024 to 1/256 of the pixels of the image. A CT image has been tampered with at four different locations, and the tampered pixels only account for about 1%. In other words, 99% of an image is interference information. The untampered part is equivalent to the "cover" of the tampered part, which seriously hinders the model from learning the difference between positive and negative images. That is why even state-of-the-art methods, like the methods based on saturation cues [5], frequency analysis [9] and spectral regularization optimization [10], are challenging to detect the tampering trace of CT-GAN. More serious is that CNN is not sensitive enough to small tampered regions, so it is difficult to detect such attacks accurately. Unfortunately, almost all current detection methods are based on deep CNN, so it is challenging to detect directly. The current methods for distinguishing whether an image is generated by GAN aim at the images wholly or mostly generated by GAN. There is no particular detection model for GAN-based small region forgery attacks in medical images like CT-GAN for the time being. In Section 4.4, We have tried to use the whole CT image as input to train the state-of-the-art network. Unfortunately, the result is inferior.

## 3 Our method

### 3.1 Motivation

Medical images are critical private information and are vitally important to the patient's life. At present, the integrity of medical images faces the threat of GAN-based small region forgery attacks. However, there is no practical method to detect GAN-based small region forgery attacks in medical images.

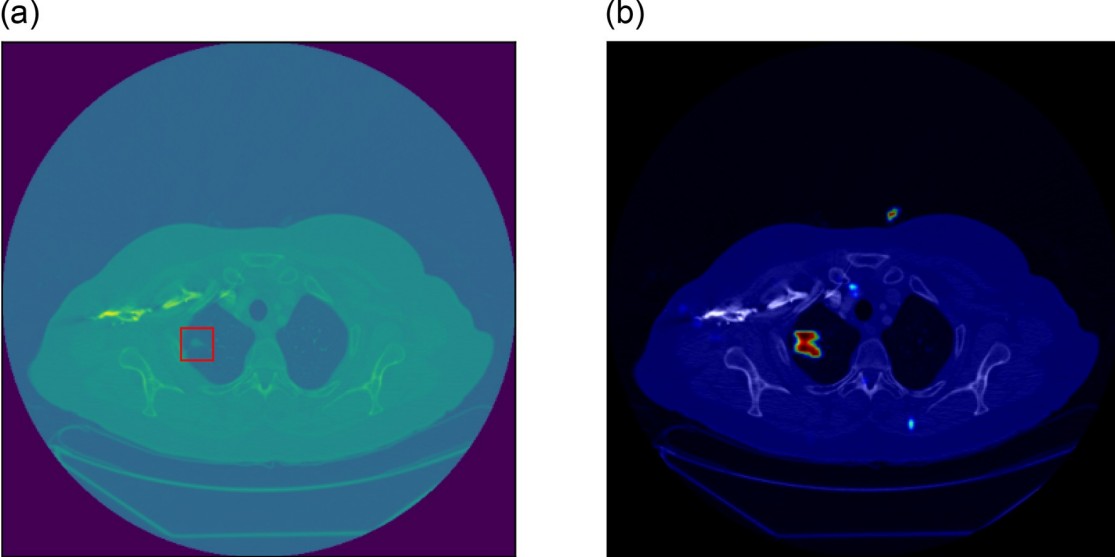

**Fig 2. Example of a CT image injected into a lung nodule.** (a) is a tampered CT image. The nodules in the red box are injected by CT-GAN. (b) is the heatmap corresponding to (a), in which the bright red spot corresponds to the injected nodule. Because the preset sliding window stride is greater than 1, the size of the heatmap is smaller than the original CT slice image. In order to facilitate observation, we enlarged the thermal map, superimposed it on the CT image, and adjusted their colors.

There are two main reasons why attacks like CT-GAN are difficult to detect. First, medical images' style, content, and storage format are very different from the normal images. If we convert these DICOM images to any other image format, it will lose information like pixels or meta-data. Therefore, the model training by normal image can not be well generalized to CT images, making the pre-training model unusable. Training a new model will need much more data. Unfortunately, the sample data of medical images is very limited due to the restrictions on the use of data concerning health under the privacy regulations like California consumer privacy act (CCPA) or general data protection regulation (GDPR). Hence, we cannot try to solve this attack from the perspective of training data. Moreover, what makes the detection task more challenging is that the tampered region is very small while the entire CT image is large. As mentioned above, the tampered region is less than ($32 \times 32$) when the entire CT is ($512 \times 512$). This means the ratio of a single tampered region in the original image may be less than 0.4%. This greatly reduces the sensitivity of general CNN detection methods since the loss of spatial information limits the learning ability of CNN. Hence, directly detecting the whole image will result in very low accuracy. Based on the above, even state-of-the-art methods are challenging to detect CT-GAN attacks.

Although no specific method can be implemented directly to detect the GAN-based small region forgery attack in medical images, some research works can still inspire us to design an effective method.

Andreas *et al.* [3] used a face tracking method to extract the face area of the image. They found that if the extracted facial information is used as the input of the detector, it will be more accurate than directly using the entire image as input. It means that the neural network can achieve a better performance if the classifier focuses on more precise regions. Following their idea, we refer to the common preprocessing method of copy-move forgery detection, making the detector pay more attention to the local part of the image through a sliding window. Specifically, we split the target CT image into many small sub-images to train a local detector with a corresponding method for using local classification results to determine global classification results.

Chollet *et al.* [39] replaced the Inception modules with depthwise separable convolutions and proposed their method named XceptionNet for computer vision. Since XceptionNet makes more efficient use of model parameters, compared to Inception V3 [43], it shows better runtime performance and higher accuracy on large-scale datasets like ImageNet while having fewer parameters than general deep CNN. This architecture can effectively reduce overfitting when we cannot collect more data. Therefore, considering these features, we designed our method inspired by the XceptionNet to detect the GAN-based local tampering attacks.

## 3.2 Threat model

In this paper, we have the below assumptions. (i) We assume that the attacker's target is the medical image. (ii) We assume that the attacker utilizes the GAN-based method instead of traditional methods like copy-move or image-splicing. (iii) We assume that the attacker realistically alters the contents of a 3D scan while considering nearby anatomy and can be completely automated.

## 3.3 Overview

As can be seen from Fig 3, our detection method is divided into two stages: local detection and global classification. The method we propose is outlined below.

In the local detection stage, small sub-images are cut out from CT slices in a planned way to train the local detector neural network. The size of the cropped sub-image is small enough for

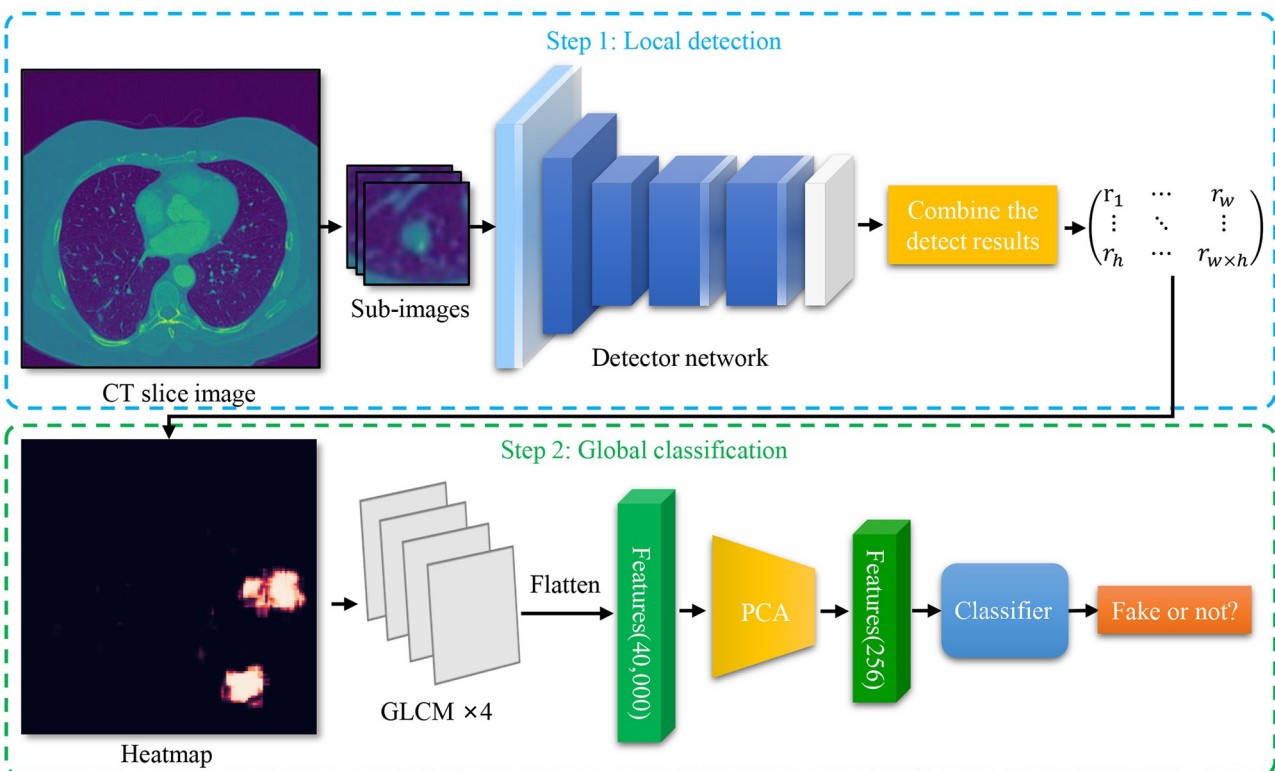

**Fig 3. Overview of our method.** We cut out small sub-images from CT slices to train the local detection neural network. Each sub-image will be detected and output a tampered probability. The detection results were combined according to the position to generate a heatmap. Then we use GLCM to extract the features from the heatmap, which are used for PCA and classifier model training. We use the trained model for global classification.

the tampered region. In this way, the authentic regions are not enough to hinder the detector. The detector can focus on learning the difference between the real image and the GAN-generated image. The tampered region may be hidden in the original image as the background when testing. Therefore, to minimize the missed judgment, our method detects each sub-image divided by the sliding window and predicts the tampered probability of each sub-image. When all sub-images were detected, the results were combined according to the position to generate a heatmap. This heatmap can intuitively reflect which region in the original image may have been tampered with by GAN. In the global classification stage, we use GLCM to extract the features from the heatmap, which are used for PCA and classifier model training. GLCM can make the features of the heatmap more prominent. We use the trained model for global classification.

Intuitively, our method allows the neural network to observe the details of the image more carefully instead of looking at the overall situation. Thus it has a better performance when facing GAN-based small region forgery attacks.

### 3.4 Local detection network architecture

Because our training data is insufficient and the hardware is not powerful, we tend to use lightweight networks as the local detector. Using depthwise separable convolution can reduce a large number of required training parameters while maintaining a good training effect. For example, XceptionNet [39] and MobileNet [44] both construct the primary part of the network

with depthwise separable convolution, and they perform well in image classification. But our classification task does not need a too-deep network. Because the training data structure is too simple, using a network like XceptionNet or MobileNet will waste many computing resources and may lead to network degradation or over-fitting. Therefore, we designed a shallower network as our sub-image classifier based on the depthwise separable convolution. Our network structure is shown in Fig 4. The network's input is a (32 × 32) image matrix, and the features of the image are extracted through a small number of traditional convolutions and a large amount of depthwise separable convolutions.

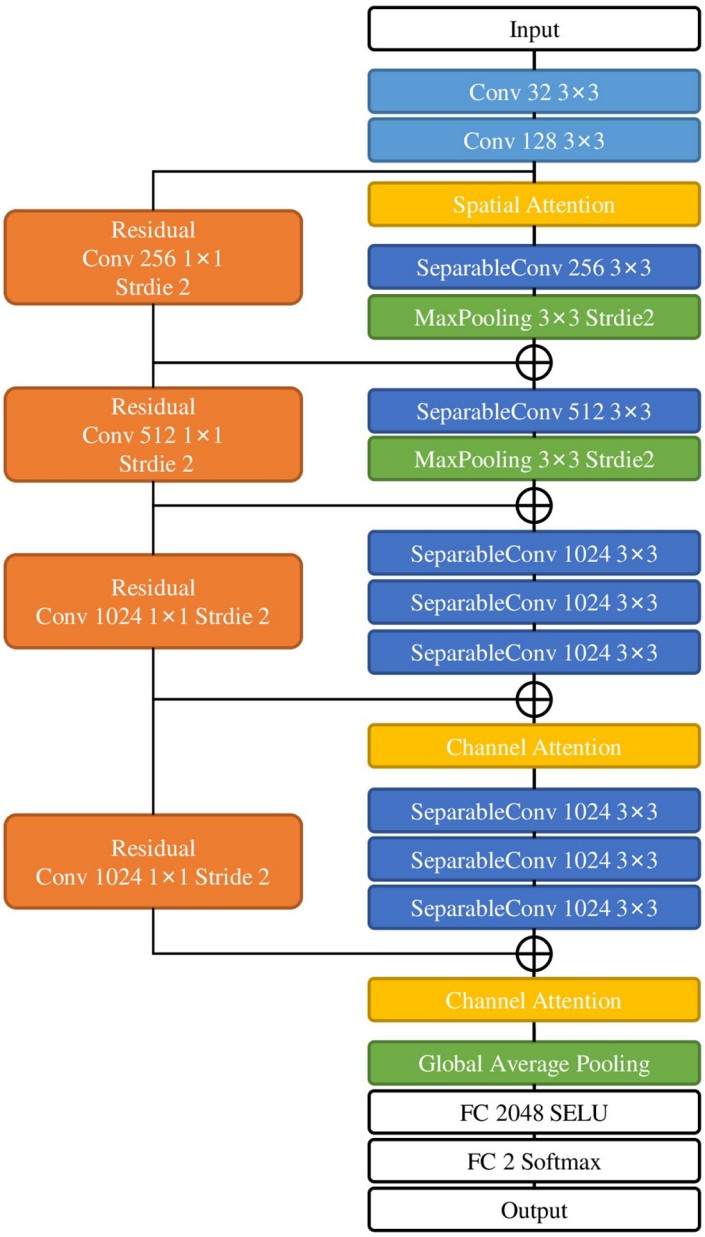

**Fig 4. The network architecture.** If there is no description, the default stride of the convolution operation is 1, the padding operation defaults to "SAME", the activation function defaults to Relu, and each convolution and depthwise separable convolution layer is followed by batch normalization by default.

The attention mechanism can effectively improve the performance of the deep learning model. The attention mechanism is often used by copy-move detection and other detail-oriented tamper detection methods. Inspired by Woo *et al.* [45], we design a simple attention mechanism for our network. Our attention mechanism is shown in Fig 5. The spatial attention and channel attention can be computed by Eqs (1) and (2). This network sets the channel attention module after the convolution blocks with the largest number of channels. It is more significant to use channel attention here. Similarly, because the pooling layer will further reduce the size of the feature image, we set the spatial attention module before the convolution block containing the pooling operation, where the feature image size is the largest. After adding the attention module, the training cost does not increase much, but it can significantly improve the global classification performance.

$$A_s(\boldsymbol{F}) = \sigma(Conv([MAX_s(\boldsymbol{F}); AVG_s(\boldsymbol{F})])) \tag{1}$$

$$A_c(\boldsymbol{F}) = \sigma(FC(MAX_c(F)) + FC(AVG_c(\boldsymbol{F}))) \tag{2}$$

The design of residual block refers to ResNet [38], and it maintains the same number of convolutional kernel channels when the input feature image size matches the output size. The number of convolution kernels and channels is doubled when the input feature image size is different from the output (through the pooling layer). In the fully connected layer of our

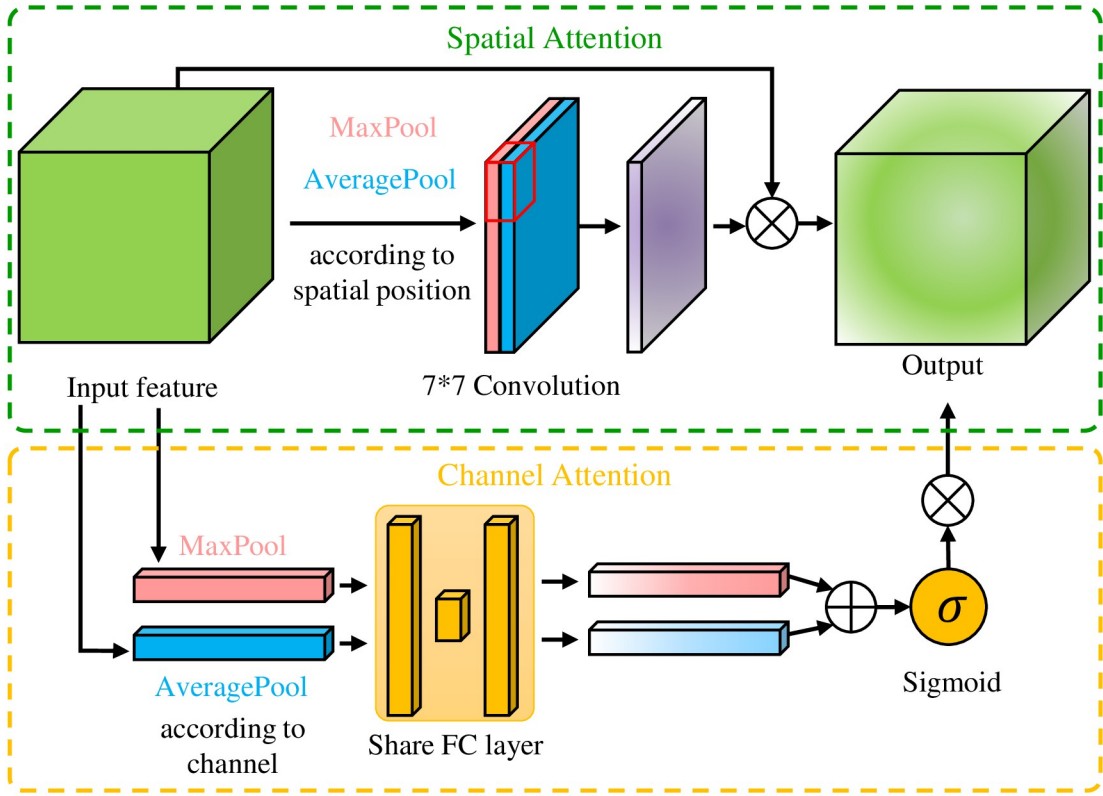

**Fig 5. The attention module of our network.** The number of nerve cells in the three FC layers of channel attention is *C*, *C*/4, *C*, where *C* means the number of channels. The stride of convolution is 1, using the SAME Padding, and the activation function is Sigmoid.

network, we employ the Selu [46] as the activation function. The Selu function is given by Eq (3), where $\lambda$ and $\alpha$ are two meticulously designed numbers. It has better performance than Relu in the full connection layer. Our network can save computing resources through the above network structure while maintaining high accuracy.

$$Selu(x) = \lambda \begin{cases} x & (x > 0) \\ \alpha e^x - \alpha & (x \leq 0) \end{cases} \tag{3}$$

## 3.5 Global classification method

In our method, the window size is fixed. This is slightly different from the sliding window commonly used in target detection tasks. In the task of target detection, if the window only covers a part of the target, the model may not be able to classify the target correctly, so it is necessary to adjust the window size and traverse the image multiple times. However, in our task, even if the window is only a part of the GAN generation region, the model can determine whether it is generated by GAN with high accuracy. Therefore, we only need a smaller window size to avoid the influence of authentic regions.

First of all, calculate a series of coordinates as the center coordinates of sub-images. Then, crop a sub-image with a size of $(32 \times 32)$ according to these center coordinates. The reason for performing the crop operation is that the practical part of the CT image corresponds only to the interior of a circle tangential to the square frame. Furthermore, most of the tampering occurs in this area. It will waste a lot of time and space if the extra part is included in the calculation.

The size of the CT image is marked as $CT_{size}$, the sub-image size is marked as $img_{size}$, and the stride marked as $s$. We calculate the longitudinal coordinates of all rows as follow:

$$y = \{\frac{img_{size}}{2} + i \times s\} \tag{4}$$

Where

$$i = 0, 1, 2, \ldots, \lfloor \frac{CT_{size} - img_{size}}{s} \rfloor \tag{5}$$

Then, for a row with $y = h$, we calculate the horizontal ordinates as follow:

$$x = \{ct_{size}/2 + j \times s\} \tag{6}$$

Where

$$j = 0, 1, 2, \ldots, \lfloor \frac{w - img_{size}}{2s} \rfloor$$

$$w = \lfloor \sqrt{(ct_{size}/2)^2 - h^2} \rfloor \tag{7}$$

Our method can record the output result of each sub-image. Then classify whether an image has been tampered with according to these results. In addition to output a final prediction result, our model can also generate a heatmap based on the results of each sub-image (see Fig 2). For the area not counted by the formulas above, we default it to have tampered with the probability of 0.

Generally speaking, attackers usually set the tampered region square or round for GAN-based small region forgery attacks. Consequently, in theory, a series of sub-images will be identified as positive after the local detection of the tampered slice. Hence, fixed patterns such as the false judgment of $n$ vertically and horizontally contiguous sub-images can be utilized to detect tampering traces of CT-GAN. Nevertheless, it is important to note that this detection method is not flawless, as the size of the tampered region in CT-GAN is variable to some extent, allowing the attacker to manipulate it. Moreover, many studies use GAN to generate more extensive and higher resolution images. Therefore, we need a flexible global classification method.

Firstly, to make the features more prominent, GLCM is used to extract the texture features in the thermal map. After rounding the local detection results (heatmap) $\times$ 100, calculate the GLCM with a distance of 1 at four angles of 0°, 45°, 90° and 135°, respectively, then get the feature matrix of $(100 \times 100 \times 4)$. Secondly, a PCA model is trained to reduce the feature to 256 dimensions. Thirdly, the feature data after dimensionality reduction are used to train an classifier model, and the best parameters of the model are found by grid search. This method can adapt to different GAN tampering region sizes.

**Algorithm 1** Generate the GLCM of heatmap

```
Input: heatmap—The heatmap matrix with size H × W. a, b—Two constants
  determined by angle and distance.
Output: GLCM—A matrix with size g × g, g is the gray levels number.
  heatmap = heatmap × 100, then round heatmap to integer, Initialize
  GLCM to 0 matrix
  x = 0, y = 0
  while x < W do
    while y < H do
      if 0 < (x+ a)<W and 0 < (y + b)<W then
        g₁ = heatmap(x, y)
        g₂ = heatmap(x+ a, y+ b)
        GLCM[g1, g2] = GLCM[g1, g2] + 1
      end if
    end while
  end while
```

## 4 Experiments

### 4.1 Implementation details

**1) Dataset and tampering methods.** The algorithms for medical image synthesis can be utilized to tamper medical images [19–24]. However, without control over what the algorithm generates, the effect of the tampering will raise suspicion. According to our thread model, if the attackers want a forgery that realistically alters the contents of a 3D scan while considering nearby anatomy and can be completely automated, they can only choose the recent variation CGAN like Pix2Pix or CycleGAN. For example, CT-GAN uses two conditional GANs to perform in-painting on 3D imagery.

For small region forgery, we use the source code of CT-GAN, train the inject and remove models with the LUNA16 data set [47]. Reichman *et al.* proposed their dataset named LuNo-Tim-CT [48] which is generated based on the LIDC/IDRI dataset. However, the quantity and quality of LuNoTim-CT is unsatisfied since it also contains copy-move and classical inpainting tampered images and is generated based on the LIDC/IDRI while the LUNA16 is much better. We then use the trained models to generate 3540 different CT scan samples. Among them, 1776 scans were injected lung cancer lesions (Equivalent to a large-diameter lung nodule), and 1764 scans were removed lung cancer lesions. For each fake sample, we select the tampered

point and two slices before and after it, five CT slices, and the corresponding five slices before tampering. In the end, 35400 CT slice images were obtained. The tampering points are the CT slices with lung nodules. Therefore, we randomly selected about half of the real CT slice images (about 8850) and replaced them with slices at random locations. Among the 35400 CT slice images, 1200 images are randomly selected as the test set, 4800 images are randomly selected as the training set of global classification, 2000 images are randomly selected as the verification set of local detection, and the remaining 27400 slices are used as the training set of local detection.

We mark the test set described in the previous paragraph as the test set CTGAN-ALL. Besides, we divide the test set CTGAN-ALL into two parts according to inject or remove tampering. The large nodule injected CT slice images, and the real large nodule images were marked as CTGAN-INJ. The large nodule removed CT images, and the real small nodule images were marked as CTGAN-REM.

In addition, eight CT scans different from the above data sets were retained. Two of them were real lung CT scans. One of them had malignant lung cancer lesions, and the other did not. These two scans were marked as MAL and BEN. In addition, one, two, and three large nodules were injected into three scans, respectively. The three scans were marked as INJ1, INJ2, and INJ3. Similarly, one, two, and three large nodules were removed from the remaining three scans marked as REM1, REM2, and REM3.

Furthermore, for the whole image forgery, we use the CycleGAN trained by LUNA16 to generate the attack dataset. We first add impulse noise and gaussian noise to 5000 CT slice images, then use CycleGAN to reduce noise. In the end, 5000 images modified by CycleGAN were obtained. The images without noise and the image denoised by CycleGAN, these 10,000 slice images are marked as the data set CycleGAN. We mark the images denoised by Cycle-GAN as the positive class and the images without noise as the negative class. Among them, 8000 images are used as the training set, 1000 images are used as the verification set, and 1000 images are used as the test set.

For each slice image in the test set, we use a $(32 \times 32)$ window to traverse the whole CT image (with the size of $(512 \times 512)$) with 4 pixels stride. Our method uses Eqs (4) and (6) to traverse the image. For each slice image of the training set and the cross-validation set, we use the method of shifting by one pixel for data enhancement so that each slice image in the training set can generate 25 sub-image images. For the fake image (positive class), we mark the coordinates of the injection center point as (0,0), take 25 coordinate points in the rectangle from (-2,-2) to (2,2). Then use these coordinate points as the center point, cut out 25 sub-images with the size of $(32 \times 32)$. For the real image (negative class), we take 10 coordinate points in the rectangle from (-2,-2) to (-1,2), and then randomly select 20 different coordinates from the coordinates calculated by Eqs (4) and (6). Taking these coordinate points as the center, and cut out 25 sub-images with the size of $(32 \times 32)$. By adding $n$ negative samples corresponding to the positive samples, the model can better learn the difference between the images before and after tampering. We found that when $n = 10$, the model's performance is better. In the experiments, the classifier model can be any algoritm. Here we choose SVM as an example.

**2) Setup.** All experiments were implemented using the Tensorflow 1.13 framework and were trained on a single NVIDIA GTX2080TI GPU. The parameters of the training phase are as follows. We set the initial learning rate to 0.0005 and use exponential decay, which decays every 600 steps and with a decay rate of 0.85. The mini-batch size is 56, the batch normalization decay parameter is 0.95, and the L2 regularization weight decay parameter is 0.0001. We use Adam optimizer to minimize cross-entropy loss. Except for the learning rate, the default parameters of the Adam optimizer are used, namely $\beta_1 = 0.9$, $\beta_2 = 0.999$, $\epsilon = 1 \times 10^{-8}$. The early stopping is set to stop training when the accuracy of the validation set no longer increases for

three consecutive epochs. If the early stop is not triggered, the training will stop after 30 epochs.

**3) Evaluation.** We regard the tampered slice image as a positive example and the real slice image as a negative example. The number of positive and negative samples in the actual scene may differ. Therefore, in addition to accuracy(ACC), we also use precision(P), recall(R) and F1-score(F1) to evaluate the model's performance. The tampering operation of CT-GAN is aimed at 3D medical images. The number of slices involved in a tampering operation can easily reach more than 30. Besides, the number of tampered slices is more if the slice interval is small. However, if the same region of 10 consecutive slices is predicted as the tampering region, we can judge that this position has been tampered with easily. However, if the above indicators are calculated in the unit of 2D slice image, the value will be deficient, which is unreasonable. Therefore, when detecting the complete CT scan, this paper also takes the 3D tampered region (a series of slice images) as the unit and counts the indicators in the following way.

For a tampered region, when 9 or more of the 10 consecutive slices, which including the tampered central slice (these slices must be positive examples in this experiment), are judged as positive examples, we consider that the tampering trace is accurately found and marked as a true positive example. Otherwise, it is regarded as a missing report and marked as a false negative example. In that case, it will be regarded as false positives. Finally, the precision, recall and F1-score are calculated in the above way.

## 4.2 Ablation study

In order to verify the effectiveness of each module in our method, we conducted ablation studies. Four experiments were used to verify the effectiveness of local detection, attention mechanism, Selu activation function, and GLCM feature extraction. In each experiment, we ablate a module from our method. In these experiments, ablate local detection is to input the complete slice image ($512 \times 512$) and use our network to train and predict directly. The experimental results are shown in Table 1.

The experimental results show that using the sliding window to divide sub-images for local detection is very helpful to detect GAN-based small region forgery attacks like CT-GAN, which significantly improves the performance of detection. Selu activation function and attention mechanism can be slightly helpful to the performance of our method. Moreover, the performance of our method can be significantly improved by using GLCM. When the above modules are used together, the improvement effect is better.

**Table 1. The ablation study result of our method.** "-SW" means that sliding windows are not used, the whole image is classified directly without local detection. "-Attention" means that attention mechanism are not used. "-Selu" means that uses Relu instead of Selu in our network. "-GLCM" means that PCA and SVM are directly used to classify the heat-map without GLCM to extract features.

| Ablated module | ACC | P | R | F1 |
|---|---|---|---|---|
| Ours-SW | 0.6583 | 0.6554 | 0.6660 | 0.6607 |
| Ours-Attention | 0.9158 | 0.9561 | 0.8717 | 0.9119 |
| Ours-Selu | 0.9192 | 0.9499 | 0.8850 | 0.9163 |
| Ours-GLCM | 0.8717 | 0.8729 | 0.8700 | 0.8715 |
| Ours | **0.9350** | **0.9628** | **0.9050** | **0.9330** |

**Table 2. The detection results of CT-GAN inject and remove attacks.** The training and testing of the two are carried out separately.

| Test set | ACC | P | R | F1 |
|---|---|---|---|---|
| CTGAN-INJ | 0.8999 | 0.9762 | 0.8200 | 0.8913 |
| CTGAN-REM | 0.9670 | 0.9937 | 0.9400 | 0.9661 |

### 4.3 Detection of CT-GAN inject or remove attack

For general GAN-based small region forgery attack, it may not use the same GAN structure to train two different models like CT-GAN. Therefore, we divided the training set into two parts according to the same way as the test set CTGAN-INJ and CTGAN-REM, then trained detector models and tested the inject and remove models of CT-GAN separately. After the training, we got two detectors for different tampering models. After that, we tested on the two kinds of tampering respectively. The test results are shown in Table 2.

The experimental results show that although training data is reduced, our model can still detect CT-GAN's inject or remove model with a high F1-score. The detection accuracy and f1-score of the inject tampering model are about 90%, while the detection accuracy and f1-score of the remove tampering model are about 97%. The above results show that our method can still effectively detect the traces of tampering in the face of a single tampering model, and our method is more sensitive to the traces of removing tampering.

### 4.4 Compare to state-of-the-art detection methods

Because other feature extraction methods for detecting GAN-generated images are not suitable for CT-GAN, as mentioned in Section 2.5, all the countermeasures listed in [41] failed to detect the lung nodule forgery. Since effective methods are all based on XceptionNet and ResNet50, we use these two most advanced deep convolutional neural networks (DCNNs) as the baseline. Specifically, some existing studies have shown that XceptionNet has an excellent performance in GAN forged image detection, and its detection accuracy can be comparable with the most advanced detection methods. Therefore, we choose XceptionNet as the baseline. In addition, because both XceptionNet and the local detection network take depthwise separable convolution as the main structure, we also select ResNet50 as another baseline. Moreover, inspired by [9] we also tried to use the DCT of the sub-image as a feature (Ours-DCT).

**1) Detect CT slices.** This experiment is divided into two kinds. One is to use the sliding window, the network as the local detector to predict the ($32 \times 32$) sub-image. The other is to train and test in a general way. The input of the network is ($512 \times 512$) complete slice images. The training set and test set are as described in Section 3. The training information is shown in Fig 6, and the test results are shown in Table 3.

Experimental results show that when using the current data set, even the most advanced DCNNs such as XceptionNet and ResNet50, the test accuracy and F1-score are only about 65%, which means that it is difficult for them to distinguish CT-GAN tampered images and real images. Their performance is improved when used as local detectors, but the performance is still unsatisfactory, which may be due to overfitting and network degradation. Our method will seriously overfit without the sliding window. However, using the sliding window, the model converges faster, and the accuracy and f1-score of our method are increased to 93%, an increase by a percentage of 28. However, all indicators have declined when using DCT as a feature, and the accuracy and f1-score are only about 86%.

**2) Detect CT scans.** In order to test the performance of our method more comprehensively, we compared our method with the latest method on complete CT scans. The model

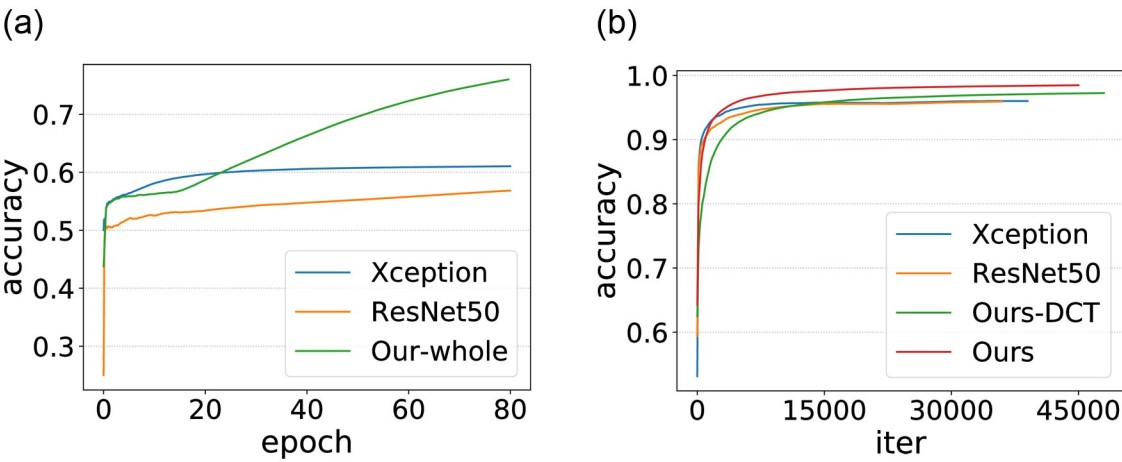

**Fig 6. The training accuracy curve.** (a) is the input with whole image (512× 512). (b) is the input with sub-image (32× 32).

used is still trained under the mixed condition of injecting and removing. The test set is the eight scans mentioned above. Fig 7 shows several continuous CT slice images near the tampering center point and the corresponding heatmap. Table 4 shows the results of our experiment.

The experimental results show that our model can effectively find the traces of CT-GAN tampering and is more stable than other methods. Our method can determine whether a scan has been tampered with automatically through a simple strategy. For example, when any *n* of *m* consecutive images are classified to be positive, it is considered that this scan has been tampered with by CT-GAN. Therefore, even CT scan is three-dimensional, while our model is two-dimensional, our model can effectively assist in distinguishing whether a CT scan has been tampered with.

In addition, for CT scans with smaller slice spacing, such as INJ1, INJ3, REM1 and REM2, our method can detect more consecutive positive samples (more than 15). However, when the slice spacing is larger, the continuous positive samples that the model can detect are fewer.

Furthermore, many misjudgments will occur in places that are unrelated to lung nodules, such as folds of clothes and calcified muscle tissue, which doctors can easily identify.

## 5 Discussion and limitations

### 1) The correlation of sub-images

In our method, we did not fully consider the correlation of sub-images since attacks like CT-GAN is a 3D medical imagery forgery. Although the current experimental results meet the

**Table 3. The detect result of CT-GAN with the state-of-the-art methods and ours.** Where "-W" means whole slice image input. "-DCT" means the local detection network is trained with the DCT features extracted from the sub-image.

| Method | ACC | P | R | F1 |
|---|---|---|---|---|
| XceptionNet-W | 0.5912 | 0.5738 | 0.7064 | 0.6332 |
| XceptionNet | 0.7125 | 0.7986 | 0.5683 | 0.6641 |
| ResNet50-W | 0.5600 | 0.5488 | 0.6690 | 0.6030 |
| ResNet50 | 0.6850 | 0.7643 | 0.5350 | 0.6294 |
| Ours-W | 0.6583 | 0.6554 | 0.6660 | 0.6607 |
| Ours-DCT | 0.8383 | 0.8512 | 0.8200 | 0.8353 |
| Ours | **0.9350** | **0.9628** | **0.9050** | **0.9330** |

(a)    (b)

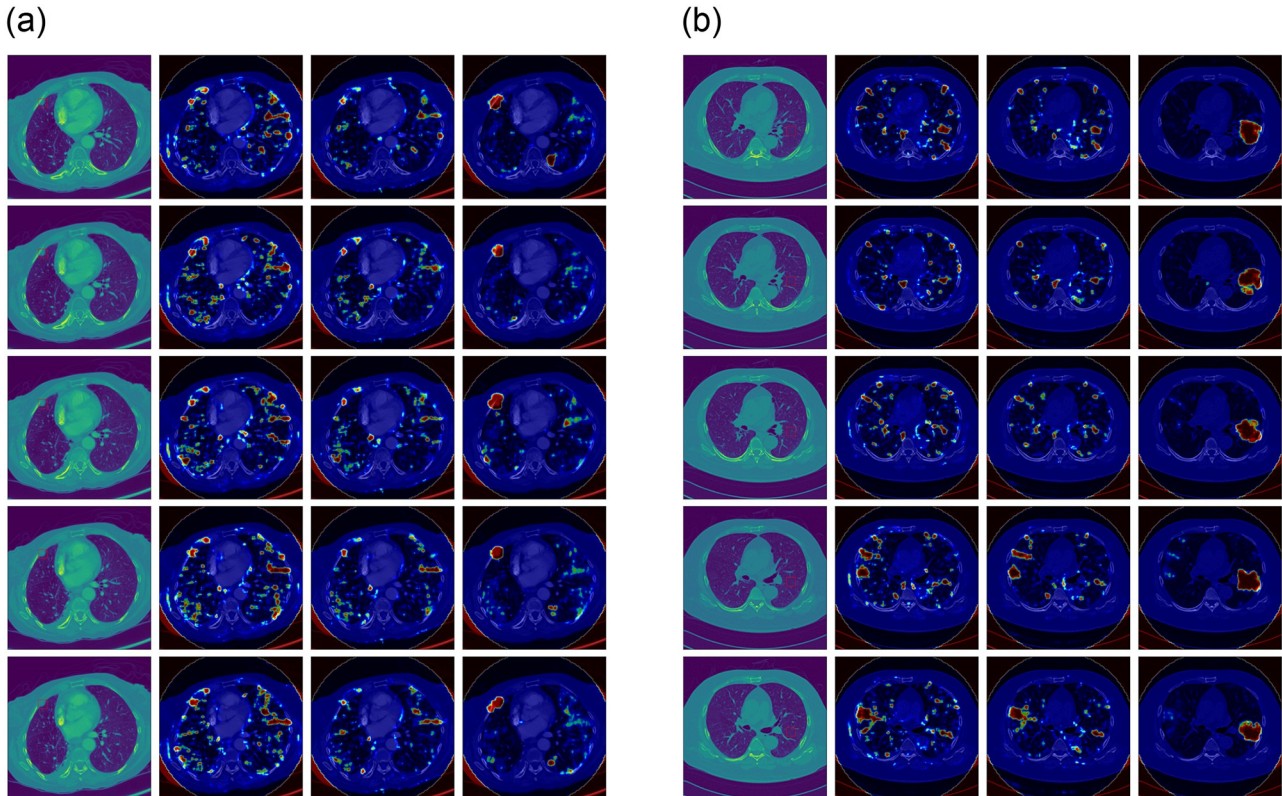

**Fig 7. A part of CT slice images and corresponding heatmaps of scans.** (a) is scan INJ1. (b) is scan REM1. The first column of each scan is the tampered CT slice image. The second column is the heatmaps output when XceptionNet is used as the local detector. The third column is the heatmaps output when ResNet50 is used as the local detector. The fourth column is the heatmaps output of our method.

requirements of the detection task, we believe that introducing this correlation into the detection task will probably help improve the detection accuracy, efficiency, or generalization. Hence, in the future, we plan to conduct more extensive experiments to find the correlation between these sub-images. For example, whether there is potentially hidden information between adjacent sub-images in the tampered region and how this hidden information helps to improve the performance.

## 2) Efficiency

Since our method divides the medical images into multiple sub-images through a sliding window, the increased number of targets would reduce the detection efficiency. With the help of a high processing speed for a single sub-image, the overall speed to detect a complete scan of the lungs is acceptable. As mentioned before, the main reason for using this sliding window is that the tampered region only occupies a small ratio of a normal image, which causes the existing detection methods that treat the target image as a whole to fail. Hence, the proposed method is the better choice from the detection point of view as it is the only detection method for the CT-GAN attack. It is worth noting that, although our method demonstrates high accuracy in detecting the tampered region when the forgery attack is applied to the entire image, its efficiency still lags behind methods that treat the target as a single unit, such as [6].

**Table 4. The detection results of complete CT scans.** The "spacing" means the spacing between two adjacent slice images. The "2D" means the indicators are calculated in the unit of a 2D slice image. The "3D" means the indicators are calculated in the unit of a 3D tampering region. There is no difference between the "2D" and "3D" methods in detection but in evaluation.

| Test set | Spacing (mm) | Method | TP | TN | FP | FN | Accuracy | Precision | Recall | F1-score |
|---|---|---|---|---|---|---|---|---|---|---|
| BEN | 2.5 | Xception | 0 | 162 | 16 | 0 | 0.9101 | - | - | - |
| | | Resnet50 | 0 | 166 | 12 | 0 | 0.9326 | - | - | - |
| | | Ours-2D | **0** | **178** | **0** | **0** | **1.0** | - | - | - |
| MAL | 0.625 | Xception | 0 | 433 | 47 | 0 | 0.9101 | - | - | - |
| | | Resnet50 | 0 | 413 | 67 | 0 | 0.9326 | - | - | - |
| | | Ours-2D | **0** | **479** | **1** | **0** | **0.9979** | - | - | - |
| INJ1 | 1.0 | Xception | 23 | 161 | 30 | 53 | 0.6891 | 0.4340 | 0.3026 | 0.3566 |
| | | Resnet50 | 12 | 176 | 15 | 64 | 0.7041 | 0.4444 | 0.1579 | 0.2330 |
| | | Ours-2D | **43** | **182** | **9** | **33** | **0.8427** | **0.8269** | **0.5658** | **0.6719** |
| | | Ours-3D | 1 | - | 0 | 0 | - | 1.0 | 1.0 | 1.0 |
| INJ2 | 2.5 | Xception | **25** | **68** | **1** | **37** | **0.7099** | **0.9615** | **0.4032** | **0.5682** |
| | | Resnet50 | 21 | 68 | 1 | 41 | 0.6794 | 0.9545 | 0.3387 | 0.50 |
| | | Ours-2D | 15 | 69 | 0 | 47 | 0.6412 | 1.0 | 0.2419 | 0.3896 |
| | | Ours-3D | 2 | - | 0 | 0 | - | 1.0 | 1.0 | 1.0 |
| INJ3 | 0.625 | Xception | 39 | 290 | 14 | 137 | 0.6854 | 0.7358 | 0.2216 | 0.3406 |
| | | Resnet50 | 32 | 291 | 13 | 144 | 0.6729 | 0.7111 | 0.1818 | 0.2896 |
| | | Ours-2D | **79** | **304** | **0** | **97** | **0.7979** | **1.0** | **0.4489** | **0.6196** |
| | | Ours-3D | 2 | - | 0 | 0 | - | 1.0 | 1.0 | 1.0 |
| REM1 | 1.8 | Xception | 11 | 88 | 2 | 55 | 0.6346 | 0.8462 | 0.1667 | 0.2785 |
| | | Resnet50 | 7 | 89 | 1 | 59 | 0.6154 | 0.8750 | 0.1061 | 0.1892 |
| | | Ours-2D | **42** | **90** | **0** | **24** | **0.8462** | **1.0** | **0.6364** | **0.7778** |
| | | Ours-3D | 1 | - | 0 | 0 | - | 1.0 | 1.0 | 1.0 |
| REM2 | 1.8 | Xception | 41 | 88 | 0 | 64 | 0.6346 | 0.8462 | 0.1667 | 0.2785 |
| | | Resnet50 | 30 | 86 | 2 | 75 | 0.6010 | 0.9375 | 0.2857 | 0.4380 |
| | | Ours-2D | **80** | **88** | **0** | **25** | **0.8705** | **1.0** | **0.7619** | **0.8649** |
| | | Ours-3D | 2 | - | 0 | 0 | - | 1.0 | 1.0 | 1.0 |
| REM3 | 2.5 | Xception | 20 | 50 | 1 | 61 | 0.5303 | 0.9524 | 0.2469 | 0.3922 |
| | | Resnet50 | 19 | 50 | 1 | 62 | 0.5227 | 0.950 | 0.2346 | 0.3762 |
| | | Ours-2D | **59** | **51** | **0** | **22** | **0.8333** | **1.0** | **0.7284** | **0.8429** |
| | | Ours-3D | 3 | - | 0 | 0 | - | 1.0 | 1.0 | 1.0 |

## 3) Generalization

The two most commonly used GAN structures in the medical image field are pix2pix and CycleGAN. CT-GAN uses the pix2pix structure, and our experiments have demonstrated the accuracy in detecting forged medical image generated by CT-GAN. On the other hand, many studies are based on the CycleGAN structure. For example, using CycleGAN to synthesize missing PET from MRI [29], learning automatic X-ray image parsing from labeled CT scan [30], automatic tumor segmentation [31], synthesized medical images [32, 33], and reconstructing CT [34]. Therefore, we chose CycleGAN to construct another data set to exam the performance of our method. The CycleGAN data set is described in Section 4.1. The medical images denoised by CycleGAN can be regarded as entirely generated by CycleGAN. The test results show that our method can classify CycleGAN tampered medical images and real medical images with 99.8% accuracy. In addition, if we do not use machine learning but use some fixed pattern for global classification, it is challenging to classify GAN-based small region forgery and the image generated by GAN wholly.

CycleGAN and pix2pix are the two most commonly used GAN structures in medical image synthesis. Our method can effectively detect the images generated by CycleGAN and pix2pix. Although our method can detect the images generated by the same or similar GAN, the detection effect of other GAN models not in the training set is not as good as the former. Many studies [2, 5, 6, 8–10, 49] want to improve the generalization ability of GAN detection methods. This may be achieved by studying the common defects of CNN or GAN. For example, Wang *et al.* [49] tested multiple latest image generation models and found that the images generated by CNN today have certain common defects. Chai *et al.* [2] summarizes which parts are likely to cause the face images generated by GAN to be recognized. However, the above studies did not take GAN-based small region forgery attacks into consideration. How to combine these studies with GAN-based small region forgery attacks is still a problem. We plan to study this in the future.

### 4) Traditional vs. GAN-based tampering

As mentioned in Section 3.2, our work focuses on detecting GAN-based tampering. That is because of three reasons.

First, traditional methods like copy-move and image-splicing are commonly used in image forgeries. By these methods, the attacker can duplicate content within the same image or from one image to another to cover up, add or modify something [50]. Many works are already focused on how to detect traditional image forgery, and these techniques can effectively identify image tampering attacks.

More importantly, both copy-move and image-splicing are performed in 2D using image software such as Photoshop. In contrast to the Photoshopping approach, for medical images, the CT scans are 3D images taking many 2D scans of the body over the axial plane (from front to back) along the body. The human body is complex and diverse in the 3D views, making it difficult to inject or remove cancers and tumours realistically since they are usually attached to nearby anatomy. Moreover, CT scanners have distinct localized noise patterns that are visually noticeable [51]. Copy-move or image-splicing will raise suspicion under the supervision of a specialist radiologist. Hence, identifying traditional medical image tampering is not a challenging problem.

It is also important to note that the tampering attacks need to automate the entire process since the radiologist will make a diagnosis immediately after performing the scan. However, the traditional tampering method can only partially be automated.

## 6 Conclusion

We propose a new method to detect GAN-based small region forgery attacks in the medical image. GAN-based small region forgery attacks under constraints targeting medical images like CT-GAN are challenging to detect by existing models that take whole images as input. We utilize our two-stage cascade framework, which uses a sliding window to train and test a light neural network in units of sub-images, then make the global classification by hyperplanes in an infinite-dimensional space. Experiments show that our method can detect SOTA GAN-based tampering traces more accurately than other detection methods under the same data set.

## Author Contributions

**Data curation:** Jianyi Zhang, Xuanxi Huang.

**Funding acquisition:** Jianyi Zhang.

**Investigation:** Jianyi Zhang.

**Software:** Xuanxi Huang, Yuyang Han, Zixiao Xiang.

**Supervision:** Jianyi Zhang.

**Validation:** Xuanxi Huang.

**Writing – original draft:** Jianyi Zhang, Xuanxi Huang.

**Writing – review & editing:** Jianyi Zhang, Yaqi Liu.

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
