## [Decision Letter · Decision Letter 0]

29 May 2023

PONE-D-23-02950GAN-based Medical Image Small Region Forgery Detection via a Two-Stage Cascade FrameworkPLOS ONE

Dear Dr. Zhang,

Thank you for submitting your manuscript to PLOS ONE. After careful consideration, we feel that it has merit but does not fully meet PLOS ONE’s publication criteria as it currently stands. Therefore, we invite you to submit a revised version of the manuscript that addresses the points raised during the review process.

We look forward to receiving your revised manuscript.

Kind regards,

Ali Mohammad Alqudah

Academic Editor

PLOS ONE

Journal Requirements:

"Yes. Supported by the Fundamental Research Funds for the Central Universities  (328202204). For Jianyi Zhang. China"

"Yes. Supported by the Fundamental Research Funds for the Central Universities  (328202204). For Jianyi Zhang. China"

"NO authors have competing interests"

6. We note that Figure 3 in your submission contain copyrighted image. All PLOS content is published under the Creative Commons Attribution License (CC BY 4.0), which means that the manuscript, images, and Supporting Information files will be freely available online, and any third party is permitted to access, download, copy, distribute, and use these materials in any way, even commercially, with proper attribution. For more information, see our copyright guidelines: http://journals.plos.org/plosone/s/licenses-and-copyright.

a. You may seek permission from the original copyright holder of Figure 3 to publish the content specifically under the CC BY 4.0 license. 

Additional Editor Comments:

The authors respond very well to reviewer comments, however, the paper still needs some editing to make it publishable. The paper has many grammatical and writing issues that made it challenging to read, and I highly encourage the authors to proofread the paper. Also, please make sure that all figures are cited correctly.

Reviewers' comments:

Reviewer's Responses to Questions

**Comments to the Author**

1. Is the manuscript technically sound, and do the data support the conclusions?

Reviewer #1: Yes

Reviewer #2: Yes

2. Has the statistical analysis been performed appropriately and rigorously? 

Reviewer #1: Yes

Reviewer #2: Yes

3. Have the authors made all data underlying the findings in their manuscript fully available?

Reviewer #1: Yes

Reviewer #2: Yes

4. Is the manuscript presented in an intelligible fashion and written in standard English?

Reviewer #1: Yes

Reviewer #2: Yes

5. Review Comments to the Author

Reviewer #1: The authors have done good work on the title “GAN-based Medical Image Small Region Forgery Detection via a Two-Stage Cascade Framework”. It will add new knowledge and new areas of research to the subject area compared with other published material.

However, i have some minor concerns:

1. The authors have inserted Fig. 1 in the introduction section, but it was not cited inside the manuscript. Kindly check the guideline of the PLOS ONE journal and perform the required amendment.

2. It would be more appropriate for the authors to define abbreviations upon first appearance in the main text such as PCA, GAN, CAD, GLCM, CCPA, GDPR and CV field; principal component analysis (PCA), generative adversarial network (GAN) computer-aided diagnosis (CAD), gray-level co-occurrence matrix (GLCM), California consumer privacy act (CCPA) and general data protection regulation (GDPR) and computer vision (CV), respectively.

3. In the section of “Challenges”, it would be more appropriate for the authors to inserted Fig. 4 immediately after the paragraph in which the figure is cited.

4. In the following sentence “That is why even state-of-the-art methods are challenging to detect the tampering trace of CT-GAN. Because of this, many methods based on statistical characteristics, such as [6, 10, 11]”, it would be more appropriate for the authors to detail the methods with their related citation.

5. In the section of “Local detection network architecture”, it would be more appropriate for the authors to inserted Fig. 5 immediately after the paragraph in which the figure is cited.

6. In the section of “Detect CT scans”, it would be more appropriate for the authors to inserted Fig. 8 immediately after the paragraph in which the figure is cited.

7. The following sentence “When the forgery attack is applied to the whole image, although we notice that our method can also detect the tampered region with high accuracy, the efficiency still cannot catch up with those methods that treat the target as a single unit, such as [7]”, seemed as incomplete sentence. Kindly check it and perform the required amendment.

8. In the following sentence “The works such as [29–34] are based on the CycleGAN structure”, it would be more appropriate for the authors to briefly indicate the works with their related citation.

9. In section of “Evaluation “, the following sentence “Similarly, for a real region, suppose 9 or more consecutive slices in the real slice are judged as positive examples, or 9 of the 10 consecutive slices are judged as positive examples”, seemed as redundant and hanged sentence. Kindly check it and perform the required amendment.

10. Moderate editing is required throughout the manuscript, for example:

a. “The design of residual block refers to ResNet [39]. The number of convolution kernels channels is unchanged when the input feature image size is the same as the output”. Moderate editing is required.

b. “At the end of our network. Selu [47] is used as the activation function in the full connection layer of the network”. Moderate editing is required.

c. “The crop operation is because the practical part of the CT image is only the inside of a circle tangent to the square frame. Moderate editing is required

d. “s. Therefore, after local detection of the tampered slice, a series of sub-images will be judged as positive in theory. Therefore, the tampering trace of CT-GAN can be detected through some fixed modes, such as the vertical and horizontal continuous n sub-images are judged to be false”. Moderate editing is required.

e. “However, this detection method is not perfect. The size of the tampered region of GAN is not fixed. The attacker can set the size of the CT-GAN tampered region to a certain extent”. Moderate editing is required.

Best regards,

Dr. Mai Abdel Haleem Abusalah

Faculty of Medical Allied Science,

Zarqa University,

Zarqa, 13110, Jordan.

Tel: +962-796862347

e-mail: ellamomo88@yahoo.com

Reviewer #2: The authors proposed a two-stage cascade framework as a solution to GAN-based medical image small region forgery detection. The authors first train the detector network with small sub-images as the input to recognize real/fake sub-images, and classify all sub-windows in a whole slice or volume to obtain the heatmap. Then a global classification is employed by extracting gray level co-occurrence matrix (GLCM) features from the heatmap and using the SVM for recognition. Experiments show that the proposed method can obtain better results. I think this manuscript can be accepted. On a less significant note, the paper has several grammatical and writing issues that made it challenging to read. I highly encourage the authors to proof-read the paper.

6. PLOS authors have the option to publish the peer review history of their article (what does this mean?). If published, this will include your full peer review and any attached files.

Reviewer #1: **Yes: **MAI ABDELHALEEM A. ABUSALAH

Reviewer #2: No

---

## [Author Response · Author response to Decision Letter 0]

17 Jul 2023

We warmly thank the reviewers for their insightful comments and constructive feedback. They have been beneficial for us to improve the quality of our paper and better communicate the core aspects and contributions of our method. We have made significant changes to the paper to address all expected requirements for the major revision. Furthermore, we have also addressed all questions and comments raised by the individual reviewers and have added explanations to clarify unclear aspects of our paper. Throughout the paper, we have sought to improve the quality of our writing.

RESPONSES TO INDIVIDUAL REVIEWS

REVIEWER # 1

1: The authors have inserted Fig. 1 in the introduction section, but it was not cited inside the manuscript. Kindly check the guideline of the PLOS ONE journal and perform the required amendment.

Authors’ response: We apologize for the oversight in not citing Fig. 1 within the manuscript. We have relocated Fig. 1 to Chapter 3.3 as it is the overview of our method and ensured proper citation according to PLOS ONE guidelines. 

2: It would be more appropriate for the authors to define abbreviations upon first appearance in the main text such as PCA, GAN, CAD, GLCM, CCPA, GDPR and CV field; principal component analysis (PCA), generative adversarial network (GAN) computer-aided diagnosis (CAD), gray-level co-occurrence matrix (GLCM), California consumer privacy act (CCPA) and general data protection regulation (GDPR) and computer vision (CV), respectively.

Authors’ response: Thank you for your comment. We have addressed your concern by defining all abbreviations upon their first appearance in the manuscript, including PCA, GAN, CAD, GLCM, CCPA, GDPR, and CV.

3: In the section of “Challenges”, it would be more appropriate for the authors to inserted Fig. 4 immediately after the paragraph in which the figure is cited.

Authors’ response: Thank you for your valuable suggestion. We have followed your advice and inserted Fig. 4 immediately after the paragraph where the figure is cited in the "Challenges" section. This modification enhances the readability and coherence of the manuscript. We appreciate your feedback. 

4: In the following sentence “That is why even state-of-the-art methods are challenging to detect the tampering trace of CT-GAN. Because of this, many methods based on statistical characteristics, such as [6, 10, 11]”, it would be more appropriate for the authors to detail the methods with their related citation.

Authors’ response: Thank you for your comment. The revised sentence now reads: ``That is why even state-of-the-art methods, like the methods based on saturation cues [6], frequency analysis [10], and spectral regularization optimization [11], are challenging to detect the tampering trace of CT-GAN." 

5: In the section of “Local detection network architecture”, it would be more appropriate for the authors to inserted Fig. 5 immediately after the paragraph in which the figure is cited.

Authors’ response: We apologize for the issue caused by the automatic typesetting in LaTeX. Based on your suggestion, we have made the necessary modification. 

6: In the section of “Detect CT scans”, it would be more appropriate for the authors to inserted Fig. 8 immediately after the paragraph in which the figure is cited.

Authors’ response: Thank you for your comment. Based on your suggestion, we have made the necessary adjustment. However, as can be seen, some of the images were automatically adjusted by LaTeX due to their size, preventing them from appearing on the same page. We will make overall modifications based on the editor's feedback during the camera-ready stage for similar cases.

7: The following sentence “When the forgery attack is applied to the whole image, although we notice that our method can also detect the tampered region with high accuracy, the efficiency still cannot catch up with those methods that treat the target as a single unit, such as [7]”, seemed as incomplete sentence. Kindly check it and perform the required amendment.

Authors’ response: Thank you for your comment. We have revised the sentence as follows: ``Although our method demonstrates high accuracy in detecting the tampered region when the forgery attack is applied to the entire image, its efficiency still lags behind methods that treat the target as a single unit, such as [7]." 

8: In the following sentence “The works such as [29–34] are based on the CycleGAN structure”, it would be more appropriate for the authors to briefly indicate the works with their related citation.

Authors’ response: Thank you for your comment. We have revised the sentence as follows: ``On the other hand, many studies are based on the CycleGAN structure. For example, using CycleGAN to synthesize missing PET from MRI [29], learning automatic X-ray image parsing from labeled CT scan[30], automatic tumor segmentation[31], synthesized medical images[32,33], and reconstructing CT[34]. " 

9: In section of “Evaluation", the following sentence “Similarly, for a real region, suppose 9 or more consecutive slices in the real slice are judged as positive examples, or 9 of the 10 consecutive slices are judged as positive examples”, seemed as redundant and hanged sentence. Kindly check it and perform the required amendment.

Authors’ response: Thank you for your comment. We have revised the sentence as follows: `` For a tampered region, when 9 or more of the 10 consecutive slices, which including the tampered central slice (these slices must be positive examples in this experiment), are judged as positive examples, we consider that the tampering trace is accurately found and marked as a true positive example." 

10: Moderate editing is required throughout the manuscript:

10.a: ``The design of residual block refers to ResNet [39]. The number of convolution kernels channels is unchanged when the input feature image size is the same as the output”. Moderate editing is required.

Authors’ response: Thank you for your comment. We have revised the sentence as follows: ``The design of the residual block is inspired by ResNet [39], and it maintains the same number of convolutional kernel channels when the input feature image size matches the output size." 

10.b: ``At the end of our network. Selu [47] is used as the activation function in the full connection layer of the network”. Moderate editing is required.

Authors’ response: Thank you for your comment. The modified sentence now reads: ``In the fully connected layer of our network, we employ the Selu [47] as the activation function." 

10.c: ``The crop operation is because the practical part of the CT image is only the inside of a circle tangent to the square frame." Moderate editing is required.

Authors’ response: Thank you for your comment. The modified sentence now reads: ``The reason for performing the crop operation is that the practical part of the CT image corresponds only to the interior of a circle tangential to the square frame." 

10.d: ``Therefore, after local detection of the tampered slice, a series of sub-images will be judged as positive in theory. Therefore, the tampering trace of CT-GAN can be detected through some fixed modes, such as the vertical and horizontal continuous $n$ sub-images are judged to be false”. Moderate editing is required.

Authors’ response: Thank you for your comment. The modified sentence now reads: ``Consequently, in theory, a series of sub-images will be identified as positive after the local detection of the tampered slice. Hence, fixed patterns such as the false judgment of $n$ vertically and horizontally contiguous sub-images can be utilized to detect tampering traces of CT-GAN." 

10.e: ``However, this detection method is not perfect. The size of the tampered region of GAN is not fixed. The attacker can set the size of the CT-GAN tampered region to a certain extent”. Moderate editing is required.

Authors’ response: Thank you for your comment. The modified sentence now reads: ``Nevertheless, it is important to note that this detection method is not flawless, as the size of the tampered region in CT-GAN is variable to some extent, allowing the attacker to manipulate it." 

REVIEWER # 2

On a less significant note, the paper has several grammatical and writing issues that made it challenging to read. I highly encourage the authors to proof-read the paper.

Authors’ response: We have thoroughly proofread the entire paper, addressing the writing issues and grammatical errors as suggested. We have made the necessary amendments to improve the readability and quality of the manuscript. Thank you for pointing out these concerns, and thank you for your comment. 

The Figure 3 in our submission contain copyrighted image. Since all PLOS content is published under the Creative Commons Attribution License (CC BY 4.0), we deleted this figure.

---

## [Decision Letter · Decision Letter 1]

7 Aug 2023

GAN-based Medical Image Small Region Forgery Detection via a Two-Stage Cascade Framework

PONE-D-23-02950R1

Dear Dr. Zhang,

We’re pleased to inform you that your manuscript has been judged scientifically suitable for publication and will be formally accepted for publication once it meets all outstanding technical requirements.

Kind regards,

Ali Mohammad Alqudah

Academic Editor

PLOS ONE

Additional Editor Comments (optional):

Reviewers' comments:

Reviewer's Responses to Questions

**Comments to the Author**

1. If the authors have adequately addressed your comments raised in a previous round of review and you feel that this manuscript is now acceptable for publication, you may indicate that here to bypass the “Comments to the Author” section, enter your conflict of interest statement in the “Confidential to Editor” section, and submit your "Accept" recommendation.

Reviewer #1: All comments have been addressed

Reviewer #2: All comments have been addressed

2. Is the manuscript technically sound, and do the data support the conclusions?

Reviewer #1: Yes

Reviewer #2: Yes

3. Has the statistical analysis been performed appropriately and rigorously? 

Reviewer #1: Yes

Reviewer #2: Yes

4. Have the authors made all data underlying the findings in their manuscript fully available?

Reviewer #1: Yes

Reviewer #2: Yes

5. Is the manuscript presented in an intelligible fashion and written in standard English?

Reviewer #1: Yes

Reviewer #2: Yes

6. Review Comments to the Author

Reviewer #1: The authors have done good work on the title “GAN-based Medical Image Small Region Forgery Detection via a Two-Stage Cascade Framework”. It will add new knowledge and new areas of research to the subject area compared with other published material.

The authors have adequately addressed all comments and performed the required amendments; hence I highly recommend accepting this interesting article.

Reviewer #2: The authors have thoroughly proofread the entire paper, addressing the writing issues and grammatical errors as suggested. They have made the necessary amendments to improve the readability and quality of the manuscript. The method might be applicable to other images as well.

7. PLOS authors have the option to publish the peer review history of their article (what does this mean?). If published, this will include your full peer review and any attached files.

Reviewer #1: **Yes: **MAI ABDEL HALEEM ABUSALAH

Reviewer #2: No

---

## [Editor Report · Acceptance letter]

17 Aug 2023

PONE-D-23-02950R1 

GAN-based Medical Image Small Region Forgery Detection via a Two-Stage Cascade Framework 

Dear Dr. Zhang:

I'm pleased to inform you that your manuscript has been deemed suitable for publication in PLOS ONE. Congratulations! Your manuscript is now with our production department. 

Kind regards, 

on behalf of

Dr. Ali Mohammad Alqudah 

Academic Editor

PLOS ONE